

# The masquerade game: marine mimicry adaptation between egg-cowries and octocorals

Juan A. Sánchez[1], Angela P. Fuentes-Pardo[2,3], Íde Ní Almhain[4], Néstor E. Ardila-Espitia[1,5], Jaime Cantera-Kintz[3] and Manu Forero-Shelton[4]

[1] Departamento de Ciencias Biológicas (Biommar), Universidad de los Andes, Bogotá, Colombia
[2] Department of Biology-Faculty of Sciences, Dalhousie University, Halifax, Canada
[3] Departamento de Biología-Facultad de Ciencias Naturales y Exactas, Universidad del Valle, Cali, Colombia
[4] Departamento de Física, Universidad de los Andes, Bogota, Colombia
[5] División de Biología Marina, ECOMAR Consultoría Ambiental, Bogotá, Colombia

Corresponding author
Juan A. Sánchez,
juansanc@uniandes.edu.co

## ABSTRACT

**Background.** Background matching, as a camouflage strategy, is one of the most outstanding examples of adaptation, where little error or mismatch means high vulnerability to predation. It is assumed that the interplay of natural selection and adaptation are the main evolutionary forces shaping the great diversity of phenotypes observed in mimicry; however, there may be other significant processes that intervene in the development of mimicry such as phenotypic plasticity. Based on observations of background mismatching during reproduction events of egg-cowries, sea snails of the family Ovulidae that mimic the octocoral where they inhabit, we wondered if they match the host species diversity. Using observations in the field and molecular systematics, we set out to establish whether the different egg-cowrie color/shape polymorphisms correspond to distinct lineages restricted to specific octocoral species.

**Methods.** Collection and observations of egg-cowries and their octocoral hosts were done using SCUBA diving between 2009 and 2012 at two localities in the Tropical Eastern Pacific (TEP), Malpelo Island and Cabo Corrientes (Colombia). Detailed host preference observations were done bi-annually at Malpelo Island. We analyzed the DNA sequence of the mitochondrial genes *COI* and *16S rDNA*, extensively used in phylogenetic and DNA barcoding studies, to assess the evolutionary relationship among different egg-cowrie colorations and morphologies.

**Results.** No genetic divergence among egg-cowries associated to different species of the same octocoral genus was observed based on the two mitochondrial genes analyzed. For instance, all egg-cowrie individuals from the two sampled localities observed on 8 different *Pacifigorgia-Eugorgia* species showed negligible mitochondrial divergence yet large morphologic divergence, which suggests that morphologies belonging to at least two sea snail species, *Simnia avena* (=*S. aequalis*) and *Simnialena rufa*, can cross-fertilize.

**Discussion.** Our study system comprised background-matching mimicry, of the masquerade type, between egg-cowries (*Simnia/Simnialena*) and octocorals (*Pacifigorgia/Eugorgia/Leptogorgia*). We observed mimicry mismatches related to fitness trade-offs, such as reproductive aggregations vs. vulnerability against predators. Despite the general assumption that coevolution of mimicry involves speciation, egg-cowries with different hosts and colorations comprise the same lineages. Consequently, we

infer that there would be significant tradeoffs between mimicry and the pursuit of reproductive aggregations in egg-cowries. The findings of this study not only contribute to the understanding of the evolution of mimicry in egg-cowries, a poorly studied group of marine gastropods, but also to the development of a new biologically meaningful board game that could be implemented as a learning tool.

## BACKGROUND

Mimicry provides some of the most spectacular outcomes of adaptation and evolution. It is usually a successful adaptation, where background matching provides concealing from predators (*Kjernsmo & Merilaita, 2012*) or prey (*Stevens & Merilaita, 2009*). However, how mimicry is achieved and maintained is not entirely understood. It should involve selection and species interactions through many generations, but the evolutionary process itself is difficult to test (*Cuthill et al., 2005*). Model species exhibiting polymorphisms leading to background mismatches, such as the peppered moth (*Van't Hof et al., 2013*), are extremely useful to fully understand the evolution and genetics of mimicry. To our knowledge, no such model for marine organisms has been well described yet.

There are many remarkable examples of mimicry and camouflage in marine animals. In fishes, for instance, there are repeated cases of mimicry, protective resemblance, and crypsis (*Randall, 2005*). Marine invertebrates such as egg-cowries (Ovulidae: Gastropoda) are recognized symbiotic snails of several kinds of cnidarians (*Schiaparelli et al., 2005*; *Reijnen, Hoeksema & Gittenberger, 2010*), whose coloration includes aposematic and camouflage patterns (*Rosenberg, 1992*; *Lorenz & Fehse, 2009*). Egg-cowries and true cowries exhibit special structures on the mantle (compound papillae), which accurately reconstruct the polyps and other structures of cnidarians (*Gosliner & Behrens, 1990*; *Schiaparelli et al., 2005*). It is presumed that part of the camouflage ability in cowries is due to developmental plasticity (*Rosenberg, 1992*), though a genetic component could also be involved, presumably very variable among the different types of cowries (*Vermeij, 2012*). The case of camouflage strategy exhibited by egg-cowries is a clear example of adaptive resemblance (*Starrett, 1993*), and it can be classified as a kind of 'masquerade.' It is a background matching mimicry where the subject is slightly different to the background but if it is only glimpsed at, it is not recognized as an edible subject different from the background (*Endler, 2006*; *Stoddard, 2012*). This is the case of the ovulids genera *Simnia* and *Simnialena* associated to octocorals of the genera *Pacifigorgia, Eugorgia*, and *Leptogorgia* at the Tropical Eastern Pacific-TEP (*Sánchez, 2013*).

Octocorals comprise some of the most conspicuous benthic organisms at rocky infralittorals throughout the TEP. Sea fans, sea whips, and sea candelabrum corals are dominant features on hard substrates in this area, reaching densities between 2 and 30

colonies m$^{-2}$ (*Sánchez et al., 2014*). Compared with only one species in the Atlantic, sea fans (*Pacifigorgia* spp.) seem to be particularly fit for the TEP, where they add up to about 35 species in tropical and subtropical waters (*Vargas, Guzman & Breedy, 2008*). Although morphological differences among species are very subtle, the color pattern at various traits comprises most of the interspecific variation (*Breedy & Guzman, 2002*). This may impose an adaptive challenge for their associated fauna, since most crustaceans and cowries seen on *Pacifigorgia* spp. match their colors (JA Sánchez, pers. obs., 2007–2012).

In this research, we studied the association between *Simnia/Simnialena* and octocorals in a system that includes three octocorals (two seafans and one seawhip) present at the oceanic island of Malpelo, Colombia (*Sánchez et al., 2012*) and over 10 sympatric sea fan and sea whip species in Cabo Corrientes, a coastal locality in Chocó, Colombia (*Sánchez & Ballesteros, 2014*), including the closely related genera of *Pacifigorgia*, *Eugorgia* and *Leptogorgia*. Our study was inspired by the remarkable accuracy of the camouflage strategy of egg-cowries inhabiting different octocoral hosts at the TEP. Are these specialized ectoparasites matching the host species diversity? In other words, does one egg-cowrie species colonize only one octocoral species, or several? Using observations in the field and molecular systematics, we tested the hypothesis whether egg-cowrie shell (color and shape) polymorphisms, that currently are categorized as distinct species, correspond to different lineages (most likely due to a coevolutionary process) or, alternatively, plasticity or any other mechanism for maintaining polymorphism in shape and coloration within an interbreeding population could explain the phenotypic diversity observed in this group. Given the mimicry specialization observed between egg-cowries and coral hosts at lower taxonomic levels (e.g., genera), data supporting this hypothesis may provide great insight on the link between micro and macro-evolution of background matching as a camouflage strategy.

The study of this rather simple but unique system may provide important data on how color polymorphisms could be retained in mimicry and may contribute to the understanding of the natural processes leading to camouflage adaptation in marine organisms. This study had two main goals: (1) the description of host preference of different color variants of egg-cowries found in different octocoral species and (2) an assessment of the phylogenetic relationship among egg-cowries with particular emphasis on those that share closely related octocoral hosts. We surveyed egg-cowries at two localities in the Colombian Pacific (Malpelo Island and Cabo Corrientes) and conducted multiyear, detailed preference observations in one of the two locations (Malpelo Island). Inspired in our observations, and in order to integrate our outreach strategy with our research, we developed a board game where evolutionary and ecological themes, such as natural selection and adaptation, are implicit. The ultimate goal of this game was to facilitate the understanding of evolutionary processes, such as adaptation and natural selection, through a ludic activity that could be easily implemented in a learning setting.

## METHODS

### Study areas

Between 2009 and 2012 using SCUBA diving we surveyed two localities at the Colombian Pacific (Fig. 1), Malpelo Island and Cabo Corrientes. Malpelo Island (4°0′N–81°36′20″W,

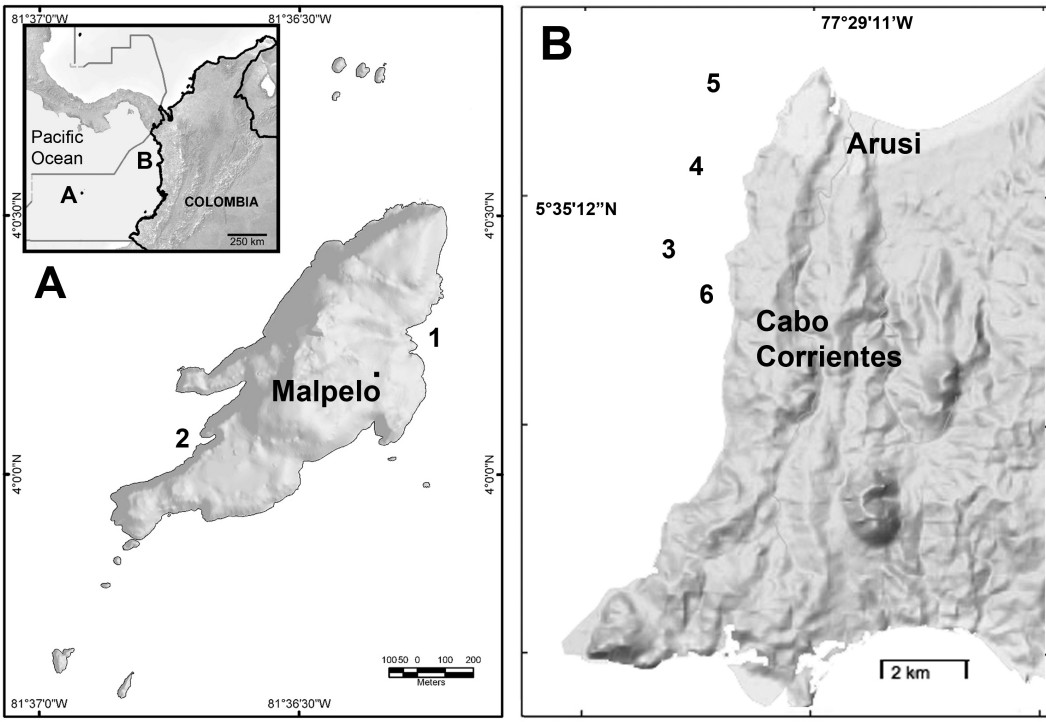

**Figure 1** **Study sites at the Colombian Pacific.** (A) Malpelo Island (1. El Arrecife; 2. La Nevera). (B) Cabo Corrientes, Chocó (3. Roñosa; 4. Piedra Bonita; 5. Parguera 7; 6. Caló).

Fig. 1A) is an oceanic rock escarpment 500 km off the continental coast of Colombia, which has been declared a conservation-dedicated national sanctuary since 1995 and a UNESCO Heritage area since 2006 (*Chasqui Velasco, Gil-Agudelo & Nieto, 2011*). There are only two sea fan species in Malpelo Island, *Pacifigorgia cairnsi* (*Breedy & Guzman, 2002*) and *Pacifigorgia* cf. *curta* (*Breedy & Guzman, 2002*), which reach an average density of 4 colonies m$^{-2}$ between 5 and 30 m depth around the island rocky littorals (*Sánchez et al., 2012*). The two species serve as a camouflage background for egg-cowries (*Sánchez, 2013*). The sea whip *Leptogorgia alba* is present in areas below 20 m and it also carries egg-cowries. As with many isolated oceanic islands, Malpelo has several endemic species (five terrestrial and seven marine) and particular ecological conditions (*López-Victoria & Werding, 2008*) that provide a unique natural experimental setting to study evolution. Endemic fish species include the Rubinoff's triplefin *Axoclinus rubinoffi* and twinspot triplefin *Lepidonectes bimaculata*, potential predators of small invertebrate such as egg-cowries (*Chasqui Velasco, Gil-Agudelo & Nieto, 2011*). We surveyed and sampled two reefs around Malpelo Island (El Arrecife and La Nevera) (Fig. 1A). The other locality studied corresponds to Cabo Corrientes, in Chocó, at the Pacific coast of Colombia (Fig. 1B). The environmental conditions in this area are quite different compared to Malpelo Island due to constant run-off from one of the most humid areas in the world, yet, the octocoral community in this rocky region is more diverse than in Malpelo (*Barrero-Canosa, Dueñas & Sánchez, 2012*). There are at least 10 octocoral species including mostly sea fans (*Pacifigorgia*) and a few species of *Leptogorgia* and *Eugorgia*, which all carry associated egg-cowries. The octocoral community has been

recently affected by the invasive snow flake coral *Carijoa riisei*, which has decimated octocoral diversity including near local extinction of some octocorals such as *Muricea* spp. (*Sánchez & Ballesteros, 2014*). In Malpelo Island as well as in Cabo Corrientes, water temperature usually does not exceed 27 °C but there is a marked upwelling season between February and April, when water temperature can be as low as 17 °C (*Sánchez et al., 2014*).

## Host preference observations

In 2008, 2009, and 2010 we surveyed four reefs at Cabo Corrientes (Roñosa, Piedra Bonita, Parguera, and Caló) (Fig. 1B). Observations and collection of egg-cowrie specimens on octocorals were carried out for at least a 30 min dive per site per year. Sampling effort and observations in Malpelo Island were more intensive. Between 2009 and 2010 two divers using SCUBA carried out egg-cowries and hosts surveys twice a year at the reefs El Arrecife and La Nevera (Malpelo Island) between 5 and 35 m depth, where permanent transects and tags for 174 *Pacifigorgia* colonies eased the biannual observations. Despite the high density of *Pacifigorgia* sea fans encountered in these surveys (*Sánchez et al., 2012*), the presence of egg-cowries and their encapsulated ovopositions were inconsistently and rarely spotted. Since some of the surveys were repeated at the same locations and depths, we realized that egg-cowries exhibit a gregarious pattern. In addition, our quantitative method, though fast and representative for surveying sea fans, was inaccurate for spotting *Simnia*/*Simnialena* egg-cowries and ovopositions given their small size and camouflage, which makes them very difficult to detect by eye in short surveys. Consequently, one more diver was added to the survey team in 2012 and 2013, whose main goal was to thoroughly search for egg-cowries on sea fans behind the divers surveying *Pacifigorgia* following the same tagged colonies. This method offered accurate information on their temporal, depth, and host preferences, including ovoposition. Most egg-masses were recorded with digital macro images (Nikon™ D7000, Nikkor micro 60 mm lens, Sea & Sea™ YS-D1 strobe and Aquatica™ AD7000 housing).

## Taxonomic identification and molecular phylogeny reconstruction of egg-cowries

We collected complete individuals of egg-cowries and tissue samples of their octocoral hosts (Research permit No. 105 (2013), Autoridad Nacional de Licencias Ambientales-ANLA, Ministerio de Ambiente y Desarrollo Sostenible, Colombia). All samples were preserved separately in 96% ethanol and stored in the laboratory at −20 °C. We obtained a merged image of the dorsal and ventral view of each snail shell using a stereo-microscope camera and the software CombineZP (last accessed on October 5, 2015, www.hadleyweb.pwp.blueyonder.co.uk). Identification of egg-cowries to the lowest taxonomic level possible was achieved following the descriptions by *Lorenz & Fehse (2009)* and *Cate (1973)*. For the octocorals, we compared the vouchers with previously identified material deposited in the Museo de Historia Natural of the Universidad del los Andes (*Sánchez et al., 2012*; *Sánchez et al., 2014*). Total genomic DNA of each specimen was extracted from about 5 mm$^2$ of tissue following a standard CTAB Phenol:Chloroform:Isoamyl Alcohol protocol (*Coffroth et al., 1992*). DNA quality was assessed in 1% agarose gel electrophoresis

in $1\times$ TBE buffer. Gels were dyed with ethidium bromide and visualized in a Gel Doc[TM]XR (Biorad, US). An approximate estimation of concentration in ng $\mu l^{-1}$ and purity (260/280 and 260/230 ratios) of each DNA sample was assessed with a NanoDrop (Thermo Scientific, US). We amplified the egg-cowries' mitochondrial genes cytochrome oxidase I (*COI*) and ribosomal large sub-unit (*16S*) using the primer pairs COI HCO-2198 (5′-TAA ACT TCA GGG TGA CCA AAA ATC A-3′) and LCO-1490 (5′-GGT CAA CAA ATC ATA AAG ATA TTG G-3′) (*Folmer et al., 1994*), and 16S-br (5′-CCG GTC TGA ACT CAG ATC ACG T-3′) and 16S-ar (5′-CGC CTG TTT ATC AAA AAC AT-3′) (*Palumbi, 1996*). PCR reactions were performed in a C1000 Thermocycler (Biorad, Hercules, CA, USA). All PCR reactions had a final volume of 15 µl including 1X buffer, 3.5 mM of $MgCl_2$, 0.2 mM dNTPs, 0,8 µg/µl of Bovine Serum Albumin, 1 µM of each primer, 1U of *Taq* polymerse, and 1–20 ng/µl of total DNA; the PCR profile started with an initial denaturation at 94 °C for 5 min, 35 cycles at 94 °C for 1 min, 44 °C for 30 s and 72 °C for 1 min, with final extension at 72 °C for 7 min. PCR products were verified in 1.3% agarose gel electroporesis in 1X TBE buffer; expected size of the amplified DNA regions were 710 bp for *COI* and 570 bp for *16S*. Contaminants remaining in the PCR products were removed following an alcohol-EDTA cleaning protocol. Sense and antisense strains of each amplified DNA region were sequenced using a Biosystems BigDye 3.1 kit and a capillary electrophoresis automated sequencer AB310 (Applied Biosystems). Raw electropherograms were checked visually using the software Geneious v4.8 (*Drummond et al., 2010*). Contigs and consensus sequences of each gene were also obtained using Geneious v4.8. We verified the overall taxonomic identity of the obtained sequences with the Basic Local Alignment Search Tool, BLAST (NCBI, US). Sequence alignment, concatenation, and phylogenetic analyses were done in Geneious v8.0.4, including the implemented packages for maximum parsimony (PAUP*), maximum likelihood (RAxML) and Bayesian inference (MrBayes), the last two analyses using the GTR model of sequence evolution as recommended by RAxML (*Stamatakis, 2015*) and default settings for getting 1,000 replicates of bootstrapping node support. A sequence from the Caribbean flamingo tongue (*Cyphoma gibbosum*: Ovulidae) was included as outgroup.

### Integrated outreach: the masquerade game
Inspired by the results obtained in this study, as well as on some elements of the classroom kits from the California Academy of Sciences, the "Coral Reef: Science and Conservation Game- The fragile coral reef (grades 3–7)" (*California Academy of Sciences, 2008*), we conceived a board game based on the mimicry adaptation of egg-cowries to coral hosts, which we think is useful for illustrating evolution and ecology concepts (see Supplemental Information for details and a game kit).

## RESULTS
### Host preference in egg-cowries
Egg-cowries in Cabo Corrientes and Malpelo Island colonized all surveyed octocoral species of *Leptogorgia, Pacifigorgia* and *Eugorgia* (Table 1 and Figs. 2–3). Large (>10 individuals) reproductive aggregations were observed on *Pacifigorgia irene*, one of the most abundant sea fans at Cabo Corrientes (Fig. 2A). Overall, all collected egg-cowries matched their mantle

Sánchez et al. (2016), *PeerJ*, DOI 10.7717/peerj.2051

**Table 1** Egg-cowrie specimen information including sample label, collection date, the putative morphologic identification, location and site of collection, depth of collection and coral host, and Genbank accession numbers for *16S* and *COI* sequences.

| Sample label | Collection date | Morphologic ID | Location | Site | Depth (m) | Coral host species | *16S* | *COI* |
|---|---|---|---|---|---|---|---|---|
| **Pacifigorgia–Eugorgia clade** | | | | | | | | |
| K136 | 19-Apr-11 | *Simnia* sp. | Cabo Corrientes | Caló | 15 | *Pacifigorgia sculpta* | KU557467 | KU557450 |
| K175 | 16-Apr-11 | *Simnia avena* | Cabo Corrientes | Roñosa | 15 | *P. stenobrochis* | KU557469 | KU557452 |
| K191 | 16-Apr-11 | *Simnia avena* | Cabo Corrientes | Piedra Bonita | 10 | *P. eximia* | KU557470 | KU557453 |
| K193 | 16-Apr-11 | *Simnia avena* | Cabo Corrientes | Roñosa | 12 | *Pacifigorgia* sp. | KU557471 | KU557454 |
| K237 | 19-Apr-11 | *Simnia avena* | Cabo Corrientes | Caló | 15 | *P. stenobrochis* | KU557472 | KU557455 |
| Y098 | 26-Feb-11 | *Simnialena rufa* | Malpelo Island | La Nevera | 10 | *P. cairnsi* | KU557473 | KU557456 |
| Y100 | 26-Feb-11 | *Simnialena rufa* | Malpelo Island | La Nevera | 10 | *Pacifigorgia* sp. *cf. curta* | KU557474 | KU557457 |
| Y101 | 26-Feb-11 | *Simnialena rufa* | Malpelo Island | La Nevera | 10 | *P. cairnsi* | KU557475 | KU557458 |
| Y109 | 26-Feb-11 | *Simnialena rufa* | Malpelo Island | El Arrecife | 15 | *P. cairnsi* | KU557476 | KU557459 |
| Y183 | 18-Apr-11 | *Simnia avena* | Cabo Corrientes | Piedra Bonita | 12 | *Pacifigorgia* sp. | KU557477 | KU557460 |
| Y185 | 18-Apr-11 | *Simnia* sp. | Cabo Corrientes | Piedra Bonita | 12 | *Pacifigorgia* sp. | KU557478 | KU557461 |
| Y188 | 18-Apr-11 | *Simnia avena* | Cabo Corrientes | Roñosa | 10 | *Eugorgia daniana* | KU557479 | KU557462 |
| Y197 | 18-Apr-11 | *Simnia avena* | Cabo Corrientes | Piedra Bonita | 12 | *Pacifigorgia* sp. | KU557480 | KU557463 |
| Y199 | 18-Apr-11 | *Simnia* sp. | Cabo Corrientes | Piedra Bonita | 12 | *P. eximia* | KU557482 | KU557465 |
| **Leptogorgia clade** | | | | | | | | |
| K116 | 17-Apr-11 | *Simnia avena* | Cabo Corrientes | Parguera 7 | 12 | *Leptogorgia alba* | KU557466 | KU557449 |
| K168 | 16-Apr-11 | *Simnia avena* | Cabo Corrientes | Roñosa | 5 | *L. ramulus* | KU557468 | KU557451 |
| Y198 | 18-Apr-11 | *Simnia avena* | Cabo Corrientes | Piedra Bonita | 12 | *Leptogorgia alba* | KU557481 | KU557464 |

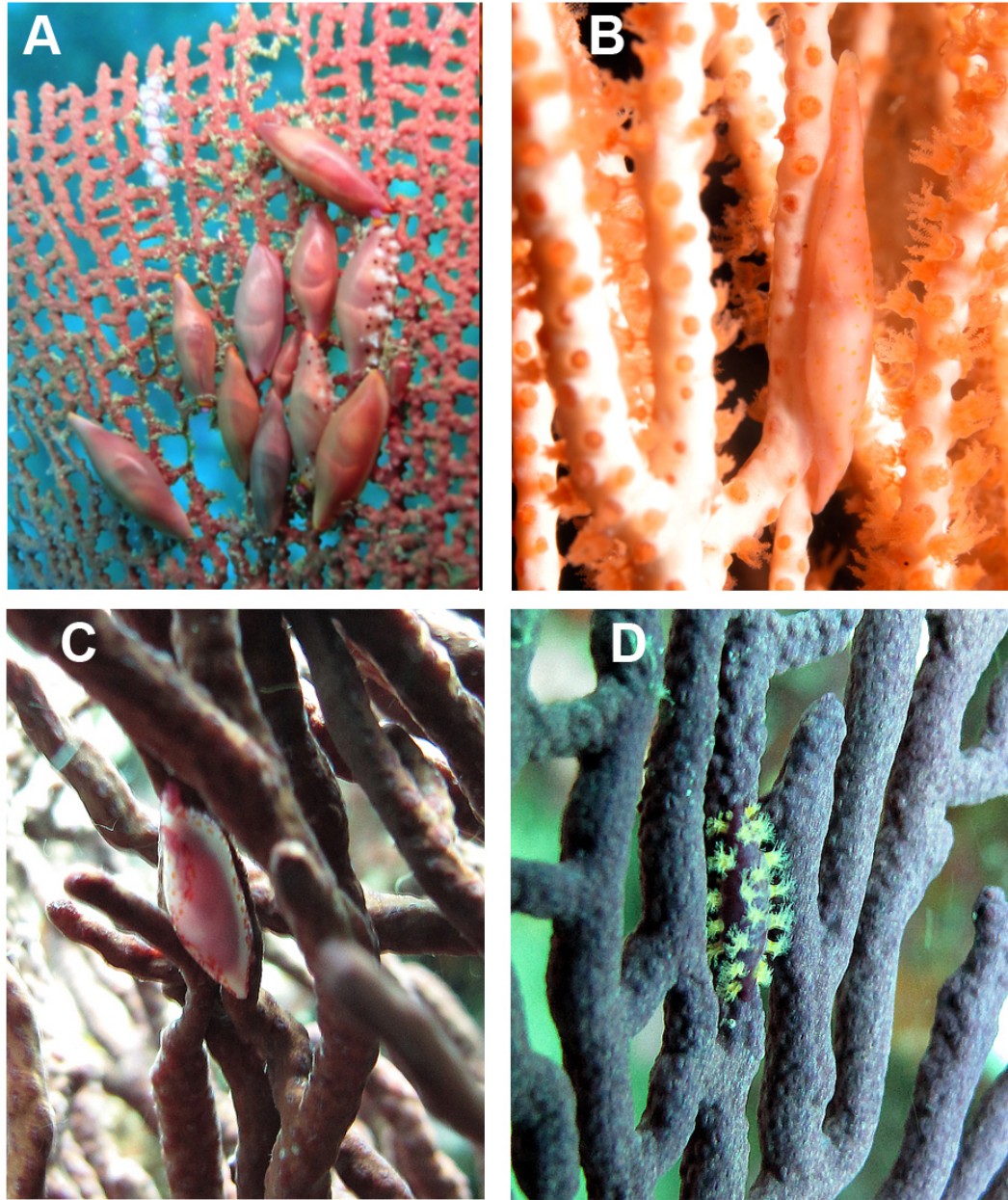

**Figure 2** **Egg-cowries (Ovulidae) observed in the Colombian Pacific (A, C and D in Cabo Corrientes, Chocó; B Malpelo Island).** Coral hosts: (A) *Pacifigorgia irene* (depth 12 m), (B) *Leptogorgia alba* (25 m) (Malpelo Island). (C) *L. ramulus* (5 m). (D) *P. stenobrochis* (15 m).

color pattern with those of the octocoral host; the shape and color of the snails associated to *Pacifigorgia* and *Eugorgia* were noticeably different respect to the snails associated to *Leptogorgia*, the first showing a red-purplish robust shell and the latter a white-pinkish elongated shell (Fig. 2).

Detailed observations at Malpelo Island revealed that about 10% of the 174 tagged colonies have at least one egg-cowrie and 4% can have an ovoposition, which corresponded to roughly 0.01 and 0.04 egg-cowries or ovopositions per square meter assuming a seafan

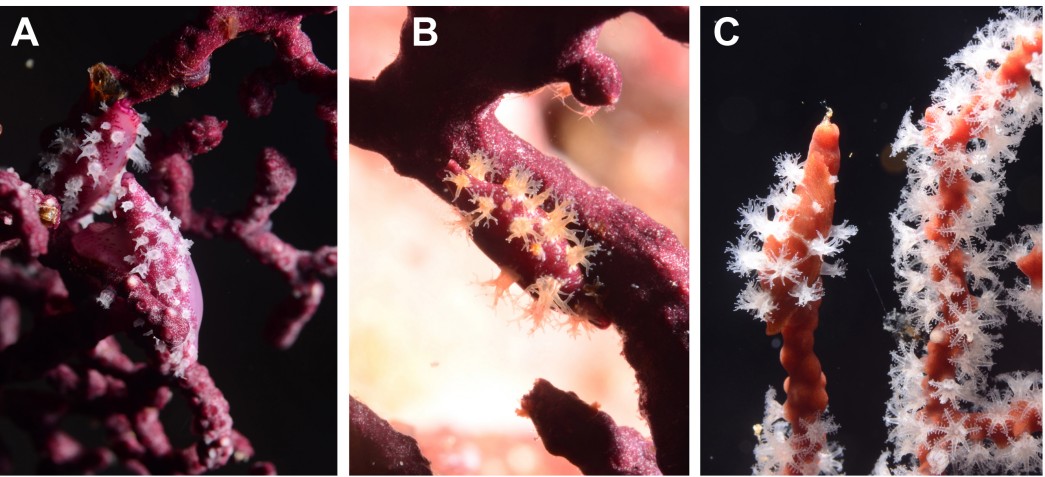

**Figure 3** **Egg-cowrie *Simnia avena* on *Pacifigorgia cairnsi* and *P.* cf. *curta* at Malpelo Island, Colombian Pacific.** (A) *S. avena* on *P.* cf. *curta*-white polyps; (B) *S. avena* on *P.* cf. *curta*-orange polyps; (C) *S. avena* on *P. cairnsi*.

density of 4 col m$^{-2}$ (*Sánchez et al., 2012*). Egg-cowries display a nearly perfect 'masquerade' background, matching the appearance of the sea fan *Pacifigorgia* by having polyp-like structures coming off the gastropod mantle (Fig. 3). There are only two species of sea fans in the infralittoral region of Malpelo Island, one has a red coenenchyme (octocoral branch tissue) with white polyps (*Pacifigorgia cairnsi*) and the other one has a blue-purplish coenenchyme with white, pink, or orange polyps (*Pacifigorgia* cf. *curta*). We observed that *Pacifigorgia* cf. *curta* egg-cowries match the background of the three polyp types present in these sea fans (Figs. 3A–3C). The polyps of the sea fans are active most of the day, but when they are not, the sea fan background changes to the color of the coenemchyme. In this case egg-cowries have to retract their mantle and the color of their shell provides mimicry. In the 174 tagged colonies, we observed less than five mismatches between shell color and coenenchyme color and those individuals usually moved towards the base of the colony. Other mismatches were observed during the reproductive period. The background matching of egg-cowries also included the color of the ovoposition, which is deposited as encapsulated eggs on sea fan branches (Figs. 4A–4C). In this trait, mismatches were commonly seen during group mating at the zone of interaction of the two seafan species (Figs. 4A–4C). This behavior could increase the chance of interbreeding among different egg-cowries species associated to a similar type of octocoral. In Malpelo Island group ovopositions were observed in July, March, and November. The two sea fan species in Malpelo Island are usually distributed at different depth ranges (*Pacifigorgia* cf. *curta* shallow, 3–10 m, and *P. cairnsi* deep, 10–30 m). We observed that snails with background mismatches were present at the boundary where the distribution of the sea fan species (*Pacifigorgia cairnsi* and *P.* cf. *curta*) coincide, or in overlapping areas where they coexist.

Given our observations of the egg-cowries natural history, we can hypothesize that background matching is under selection because there are many potential natural predators that can take advantage of mismatches. For instance, hawkfishes, predators of small invertebrates (*Froese & Pauly, 2015*), were frequently seen near sea fan colonies with

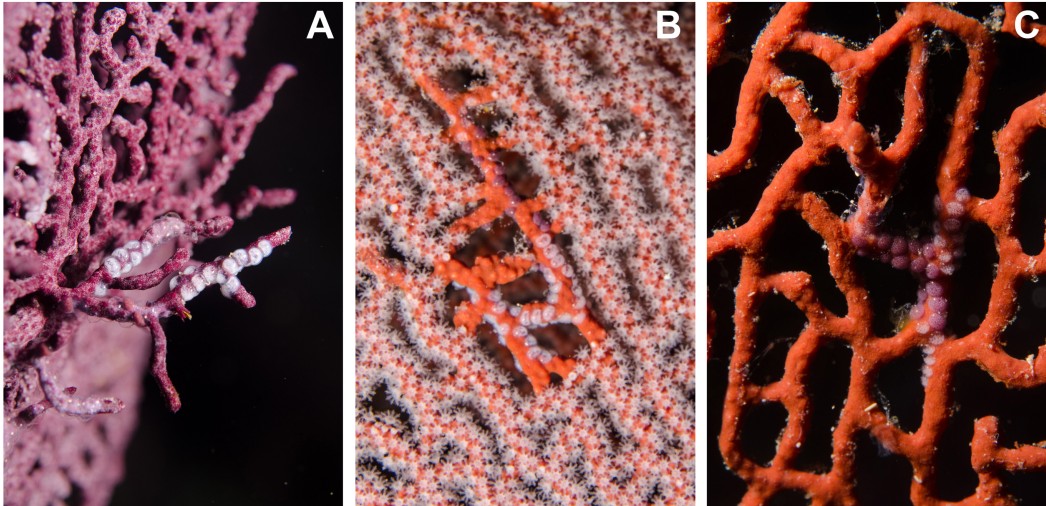

**Figure 4** **Mixed ovopositions (white and purple) left by the cowrie *Simnia avena* on sea fans at Malpelo Island, Colombian Pacific.** (A) Mixed ovoposition on purple background, *Pacifigorgia* cf. *curta*; (B–C) mixed ovoposition on red background, *P. cairnsi*.

egg-cowries (Figs. 5A–5B). The longnose hawkfish (*Oxycirrhites typus*) is adapted to hunt small invertebrates found on octocoral and black coral branches and comes in close contact with egg-cowries as it also settles on sea fans (Fig. 5A). The coral hawkfish (*Cirrhitichthys oxycephalus*) also patrols for small prey on the substrate in close contact with sea fans (Fig. 5B). Although these fish do not have a fixed territory, they spend enough time around sea fans to spot and take advantage of mismatched cowries.

Our observations suggest that color variants of egg-cowries could move freely throughout the two *Pacifigorgia* species in the island. Sea fans of these two species were as near as 1–5 mm from each other at their overlapping depth interval (4–10 m). This implies that egg-cowries could go from one colony to another without leaving their hosts. Given the abundance of predators, there should be a trade-off between reproduction and protection against predation in egg-cowries.

We observed additional threats to egg-cowries survival related to habitat destruction, i.e., sea fan mortality. During the years of this survey (2009–2012) two sources of sea fan mortality were noted in Malpelo Island, fungal diseases and an invasive coral overgrowth. Mass mortalities were observed during 2009 and 2010, sometimes reaching 70% of sea fans (*Sánchez et al., 2012*). Affected colonies had the epizootiology of the fungal disease aspergillosis (Fig. 5C) that has been recently detected elsewhere in the TEP (*Barrero-Canosa, Dueñas & Sánchez, 2012*). At two locations in Malpelo Island, 'submarino' (western most point of 'La Nevera') and some isolated rocky islets off the main island ('La Catedral'), large infra-littoral areas are completely covered by the snowflake coral *Carijoa riisei* (Fig. 5D). This is an invasive octocoral presumably brought from the Western Atlantic, which overgrows and kills sea fans in Malpelo Island and elsewhere in the TEP (*Sánchez & Ballesteros, 2014*). Though egg-cowries were seen on sea fan colonies affected by these two stressors, most of them were observed on healthy ones.

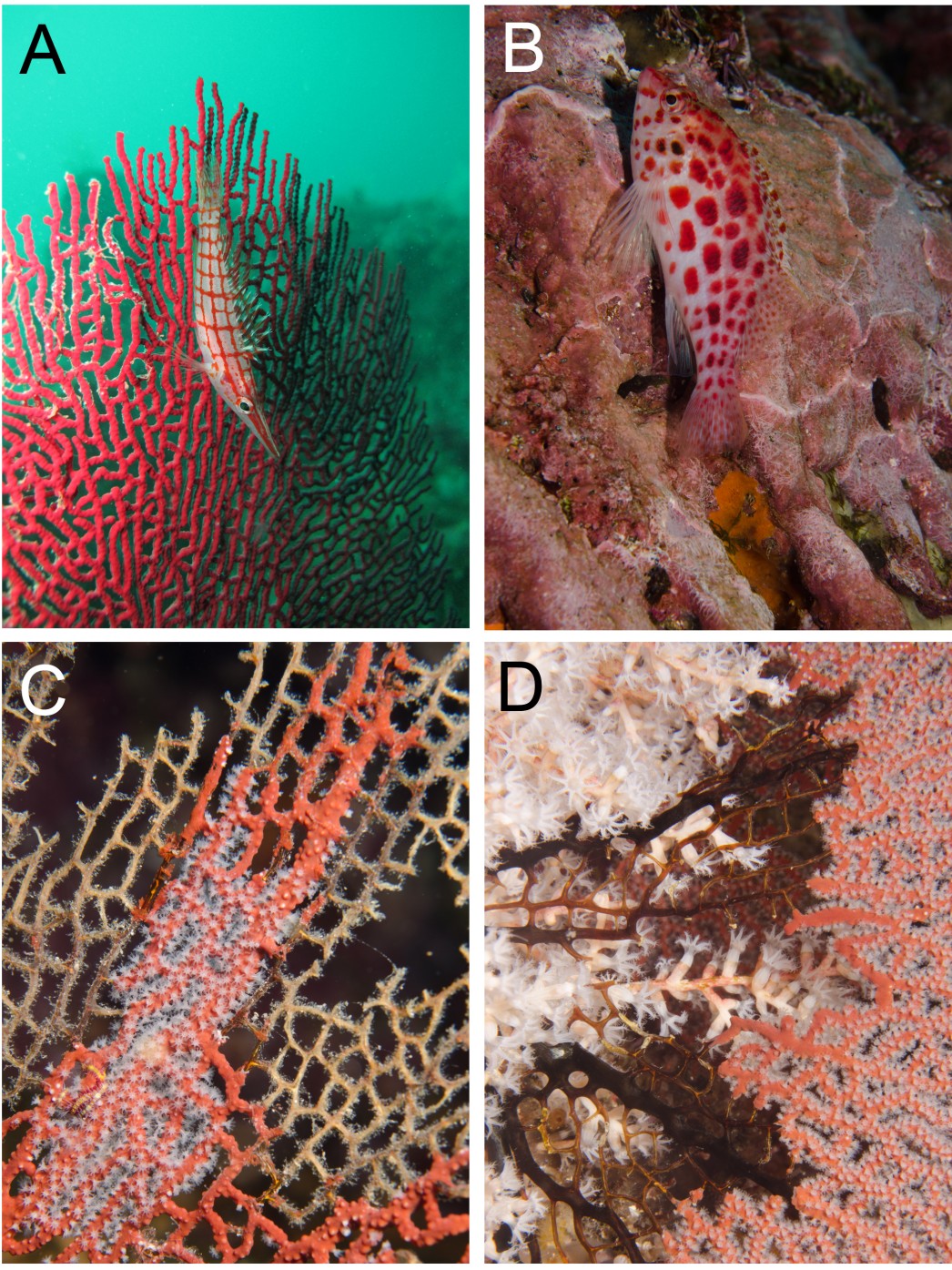

**Figure 5** **Potential threats to *Simnia* egg-cowries at Cabo Corrientes and Malpelo Island, Eastern Tropical Pacific, Colombia.** (A) Longnose hawkfish *Oxycirrhites typus* with a sea fan-like background camouflage (Cabo Corrientes); (B) Coral hawkfish *Cirrhitichthys oxycephalus;* (C) Diseased sea fan *Pacifigorgia cairnsi;* (D) The invasive snowflake coral, *Carijoa riisei*, overgrowing *P. cairnsi* (B–D Malpelo Island).

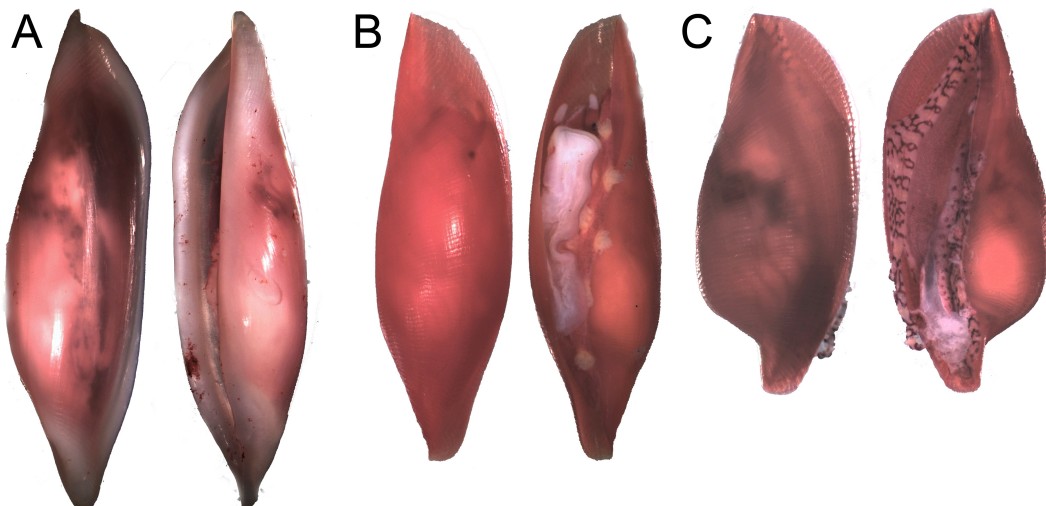

**Figure 6** **Distinct shell morphologies found in egg-cowries at the Colombian Pacific.** (A) *Simnia avena*, adult specimen K168 found on *Leptogorgia ramulus* at Cabo Corrientes, 12.6 mm (max length). (B) *Simnia avena*, juvenil specimen K191 found on *Pacifigorgia eximia*, Cabo Corrientes, 5.8 mm. (C) *Simnialena rufa*, Y100, on *Pacifigorgia* cf. *curta*, Malpelo Island, 4 mm.

## Morphologic and molecular identification

Egg-cowrie surveys revealed a species complex including the nominal species *Simnia avena* = *S. aequalis* (G.B. Sowerby II, 1832) fide *Lorenz & Fehse (2009)* and, *Simnialena rufa* = *Simnia inflexa* (G.B. Sowerby II, 1832) fide *Cate (1973)* and *Lorenz & Fehse (2009)*. The *S. avena* morphology was observed on most *Pacifigorgia*, *Leptogorgia* and *Eugorgia* at Cabo Corrientes, whereas *Simnialena rufa* was observed on *Pacifigorgia* sea fans in Malpelo Island (Table 1). Figure 6 shows the typical morphology of the two nominal egg-cowrie species observed in the Colombian Pacific. In general we encountered challenges in the taxonomic identification of egg-cowries, either because there is not a unified and updated taxonomic key for this group or due to the presence of intermediate morphologies. For instance, egg-cowries in Malpelo Island clearly looked like *S. rufa* when found on *Pacifigorgia* cf. *curta* and more like *S. avena* on *P. cairnsi*.

A phylogenetic analysis based on the sequence of the mitochondrial genes *COI* and *16S* showed two well supported clades for all the surveyed egg-cowries (Fig. 7). One clade comprised *S. avena* specimens associated to *Leptogorgia* spp and the other one included egg-cowries found on *Eugorgia* and *Pacifigorgia*, both in Cabo Corrientes and Malpelo Island (Table 1). Within each clade there was neither well-supported sub-clades nor significant sequence divergence (Fig. 7). This result supports the assumption of an important role of developmental plasticity on the variation of camouflage patterns within egg-cowries species.

## DISCUSSION

Egg-cowries at the Colombian Pacific exhibit a remarkable camouflage strategy, in a masquerade fashion, mimicking over a dozen of octocoral hosts. The negligible genetic divergence observed within clades, that included multiple species and occasionally genera,
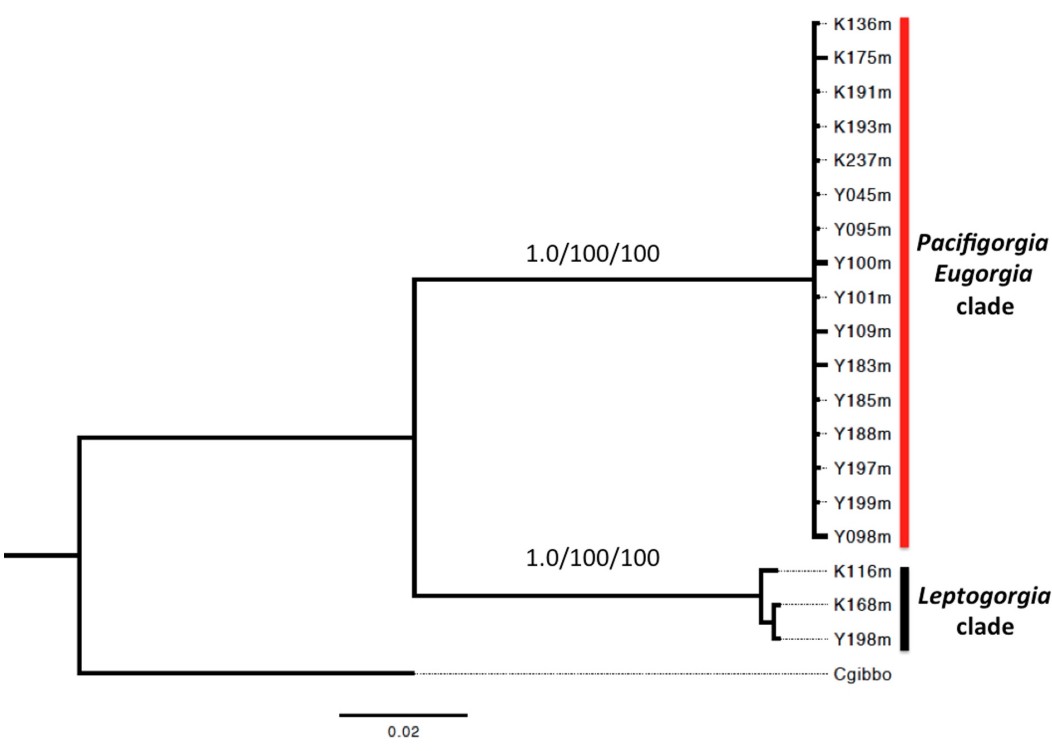

**Figure 7** **Bayesian inference phylogram using concatenated 16S and COI mitochondrial sequences.** Above node support are presented for 1,000-replicates bootstrapping values >0.7 (Bayesian posterior probabilities) and >70% maximum parsimony/maximum likelihood.

suggests that developmental plasticity should play an important role on the determination of their delicate masquerade camouflage. Though the remarkable mimicry of egg-cowries has been previously studied, we found no former record on background matching for the encapsulated eggs nor the observed oviposition mismatches at the zone of interaction of similar sea fan hosts. The main contribution of our natural history observations relies on the recognition of tradeoffs between mimicry and the pursuit of reproductive aggregations in egg-cowries.

Previous phylogenetic studies in Ovulidae provide a good framework to situate the evolutionary tempo among TEP egg-cowries. At a large phylogenetic scale, including several Ovulidae genera, cowries show some degree of host specialization within Anthozoa yet several morphologic traits used in taxonomy are polyphiletic (*Schiaparelli et al., 2005*). Likewise, there is a marked phylogenetic split between Indopacific and Atlantic Ovulidae; still, many groups at each ocean lacked phylogenetic differentiation (*Reijnen, Hoeksema & Gittenberger, 2010*; *Reijnen, 2015*). Consequently, the paucity of genetic divergence among egg-cowries from closely related octocorals, as seen in our results, is not surprising.

Mitochondrial DNA genes, such as *COI*, have been extensively used for the taxonomic identification of mollusks, including ovulids, from different parts of the world (*Layton, Martel & Hebert, 2014*; *Borges et al., 2016*). For cowries, sister group of egg-cowries, *COI* barcoding has shown a species identification error between 4 and 17% (*Meyer & Paulay, 2005*). Whereas for the egg-cowries, *Crenavolva* spp., *COI* and *16S,* the same mtDNA genes

used in this study, provided enough support for species differentiation and reviewing of the taxonomy of the genus (*Reijnen, 2015*). The modest phylogenetic divergence of the studied egg-cowries suggests a key role of phenotypic plasticity in the morphological variation.

Among the consequences of highly specialized parasitism and mimicry of the masquerade type is coevolution with their host. The case of egg-cowries is a nearly perfect masquerade to their octocoral hosts. Just in Malpelo Island there are at least four different color patterns in sea fan polyps that were matched by the egg-cowries. How did the same interbreeding population achieve this color variation? Our observations suggest that reproductive aggregation is a priority over concealment from predators and more importantly, regardless of the mates' color. This behavior is similar to what has been seen in the aposematic ovulid *Cyphoma gibbosum* for chosing among octocoral hosts (*Nowlis, 1993*). In addition, this supports the notion that conservation of polymorphisms in this trait would promote faster adaptation. Isolation with gene flow between different colorations of egg-cowries could contribute to maintain color polymorphism, which would promote faster background matching adaptation (*Gray & McKinnon, 2007*). However, the question of how much of the camouflage ability is due to developmental plasticity in response to the environment (host) and how much is due to adaptive genetics (*Rosenberg, 1992*) remains unsolved.

The link between phenotypic plasticity and diversification processes remains one of the major questions in evolutionary biology (*West-Eberhard, 2003*; *Fitzpatrick, 2012*). Phenotypic plasticity provides the adaptive canvas for further adaptation and speciation. Yet, could phenotypic plasticity promote ecological speciation? Phenotypic plasticity could impede diversification since a single genotype is supposed to give rise to different phenotypes, as we observed in egg-cowries associated to sea fans. We assume phenotypic plasticity is allowing egg-cowries to colonize many hosts in the TEP, for which specialization resembles adaptive divergence or even radiation (*Pfennig et al., 2010*). Given the potential occurrence of mating among different egg-cowrie morphotypes, a detailed analysis of their adaptive genetic variation using more powerful molecular approaches would be necessary. We consider that the *Simnia/Simnialena* complex may constitute an ideal marine system to study and test this question.

## CONCLUSIONS

Our study system comprised background-matching mimicry, of the masquerade type, between egg-cowries (*Simnia/Simnialena*) and octocorals (*Pacifigorgia/Eugorgia/Leptogorgia*). The ovoposition of the different egg-cowrie color variants also matches the host color. Egg-cowries with different color patterns but associated to similar octocoral hosts can indistinctively gather for reproductive aggregations in Malpelo Island, which was consistent with their negligible phylogenetic divergence. Egg-cowries show background mismatches in ovoposition, which constitute a particular event that could help to understand how selection operates to refine mimicry traits and promote adaptation. These novel observations inspired us to develop a biologically meaningful game that could facilitate the teaching and learning process of ecology and evolution in the classroom as well as in outreach activities, while increasing the awareness and connection of students with their environment. We invite the readers to play and share this game, available in Supplemental Information.

## ACKNOWLEDGEMENTS

We acknowledge the 2009–2012 expeditions to SFF Malpelo Island organized and supported by Fundación Malpelo and Parques Nacionales Naturales (SFF Malpelo), Colombia (Sandra Bessudo and Nancy Murillo). Special thanks to Ciaran Smyth for fruitful insights into the game and for help in testing the early versions. We appreciate the help from colleagues and students from BIOMMAR for assistance in the field and laboratory: Luisa Dueñas, Catalina Ramírez, Diana Ballesteros, Carlos E. Gómez, Fabio Casas, Lina Gutierrez, Elena Quintanilla and Dairo Escobar, among others. We appreciate the comments from our reviewers, Joana Robalo and an anonymous reviewer that greatly improved the manuscript. We recognize the participation and support from local communities in Cabo Corrientes.

### Funding

This study was funded by the Vicerrectoria de Investigaciones, Universidad de los Andes (Programas de investigación), National Geographic Society-Waitt grants, COLCIENCIAS (grant No. 1204-521-29002), and partial support of the Vicerrectoría de Investigaciones de la Universidad del Valle and the program Jóvenes Investigadores e Innovadores (APF-P) of the Department of Science, Technology, and Innovation—COLCIENCIAS. The funders had no role in study design, data collection and analysis, decision to publish, or preparation of the manuscript.

### Grant Disclosures

The following grant information was disclosed by the authors:
Vicerrectoria de Investigaciones, Universidad de los Andes (Programas de investigación).
National Geographic Society-Waitt grants.
COLCIENCIAS: 1204-521-29002.
Vicerrectoría de Investigaciones de la Universidad del Valle.

### Competing Interests

The authors declare there are no competing interests.

### Author Contributions

- Juan A. Sánchez conceived and designed the experiments, performed the experiments, analyzed the data, contributed reagents/materials/analysis tools, wrote the paper, prepared figures and/or tables, reviewed drafts of the paper.
- Angela P. Fuentes-Pardo conceived and designed the experiments, performed the experiments, analyzed the data, wrote the paper, prepared figures and/or tables, reviewed drafts of the paper.
- Íde Ní Almhain and Néstor E. Ardila-Espitia conceived and designed the experiments, performed the experiments, wrote the paper, prepared figures and/or tables, reviewed drafts of the paper.

- Jaime Cantera-Kintz and Manu Forero-Shelton conceived and designed the experiments, performed the experiments, contributed reagents/materials/analysis tools, wrote the paper, prepared figures and/or tables, reviewed drafts of the paper.

**Field Study Permissions**

The following information was supplied relating to field study approvals (i.e., approving body and any reference numbers):

Research permit No. 105 (2013) Autoridad Nacional de Licencias Ambientales-ANLA, Ministerio de Ambiente y Desarrollo Sostenible, Colombia.

Contrato de Acceso a Recursos Genéticos para Investigación Científica Sin Interés Comercial No 106, 20 August, 2014, RGE0114.

**DNA Deposition**

The following information was supplied regarding the deposition of DNA sequences:

Genbank numbers in Table 1.

**Data Availability**

The research in this article did not generate any raw data (other than DNA sequences in Table 1 and underwater images as figures).

**Supplemental Information**

Supplemental information for this article can be found online at http://dx.doi.org/10.7717/peerj.2051#supplemental-information.

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
