# Peer review of "The masquerade game: marine mimicry adaptation between egg-cowries and octocorals"

_PeerJ, doi:10.7717/peerj.2051_

## Round 0.1 · original submission · Major Revisions

The reviewers found the study of interest, however major revision have been requested before the manuscript can be considered for publication. Concerns were raised regarding the lack of description of methods, clarity of hypotheses, and connection between proposed and addressed/achieved aims. I recommend the authors address carefully all the comments and suggestions provided by the reviewers.

·

Basic reporting

no comments

Experimental design

This study lacks an evident hypothesis, perhaps relating polymorphism with octocoral diversity in both sampling sites. Although not always imperative the hypotheses help us to clarify out main aims and results and this manuscript does not follow a straightforward line of reasoning, making it difficult to follow by the reader.

Validity of the findings

Although the study provides interesting observations on background-matching mimicry the methodology used and the results obtained do not allow a clear interpretation trend resulting, in my view, in a rather speculative manuscript.

Additional comments

The manuscript “The Masquerade Game: marine mimicry adaptation between egg-cowries and octocorals” (#8738), written by Sánchez and colleagues aims to describe the marine mimicry adaptation present in egg-cowries (genus Simnia/Simnialena) and octocorals (genus Pacifigorgia/Eugorgia/Leptogorgia) from Tropical Eastern Pacific (TEP) in Colombia (Malpelo Island and Cabo Corrientes).

In my view this manuscript adds useful information to this thematic, but I have some concerns and, in my view, there is still room to some major improvements.

My major concerns are the following:

- This study lacks an evident hypothesis, perhaps relating polymorphism with octocoral diversity in both sampling sites. Although not always imperative the hypotheses help us to clarify out main aims and results and this manuscript does not follow a straightforward line of reasoning, making it difficult to follow by the reader.
- Although the study provides interesting observations on background-matching mimicry the methodology used and the results obtained do not allow a clear interpretation trend resulting, in my view, in a rather speculative manuscript.
- Considering “Egg-cowries distribution and natural history”, there are only results from Malpelo Island? If that is true please clarify all the text accordingly because it is a diferent manuscript…
- although the board game is for sure a wonderful learning tool I do not think it should be included in the manuscript as an important result. I would remove it. At maximum include it as supplementary material or add an hyperlink.
- the genes chosen are conservative mitochondrial markers and COI has been used for barcoding in several animal groups. Apparently they do not discriminate between the species present and recognized by morphological characters. Other genes should perhaps be used to assess the genetic diversity present in different ecological contexts.




Smaller concerns:

Line 100: “two sea fan species present at the oceanic island of Malpelo, Colombia (Sánchez et al., 2012), to over 10 sympatric sea fan and sea whip species in Cabo Corrientes” – if you consider sea fans and sea whips you should include the sea whip present in Malpelo also.

Line 112: “We surveyed cowries at two localities in the Colombian Pacific
and conducted detailed natural history descriptions in one of the two locations based on multiyear qualitative observations. “ - It is not clear if both locations are subjected to detailed natural history descriptions, please clarify.

Line 133 - “As with many isolated oceanic islands, Malpelo has several endemic species (5 terrestrial and 7 marine) and unusual ecological conditions (López-Victoria & Werding, 2008), providing a unique natural experimental setting to study evolution.” To describe accurately the interest of the sampling site you could detail the taxa which are endemisms (or at least the groups of organisms) and also the unusual ecological conditions found in the island…the same should be apllied to Cabo Corrientes. At general the text about sampling sites would benefit from a better description.

Line 141- “There are at least 10 species of sea fans (Pacifigorgia) and a few species of Leptogorgia and Eugorgia, which all carry associated egg-cowries.” – There is a lack of consistency in the numbers of species considered, please check the text for this.

Line 164 – “Most observations were recorded with digital macro images (NikonTM D7000, Nikkor micro 60 mm lens, Sea & SeaTM YS-D1 strobe and AquaticaTM AD7000 housing).” And the objective was?

Line 168 – “We collected complete individuals of egg-cowries and tissue samples of their octocoral hosts. All samples were preserved separately in 96% ethanol and stored in the laboratory at -20 ºC.” Why collect tissue samples from octocorals if they were not sequenced?

Line 194 – “16S and 710 bp for COI.” Please invert the order for consistency, throughout the text.

Line 201 – “miscellaneous phylogenetic analyses were done in Geneious v8.0.4, including maximum parsimony, maximum likelihood and Bayesian inference, the last two analyses using the GTR model of sequence evolution and default settings for getting 1000 replicates of bootstrapping node support.”
Please detail all analysis done and delete miscellaneous…
Why was the GTR model chosen?

Line 208- “Development of board game.” As stated above I would delete this section.

Line 225 – “Cabo Corrientes and Malpelo Island” – Please invert the order for consistency, throughout the text.

Minor suggestions:

- Abstract, Methods – octocoral hosts? DNA sequences?
- Abstract, Results – Why does the board game appear here? Is this an output?
- Abstract, Discussion - we infer there would?
- Keywords – hawkfish. Why?
- Line 81: involved, presumably very variable among the…
- Line 112: We surveyed cowries at two localities in the Colombian Pacific ((Malpelo Island and Cabo Corrientes)….
- Line 239: “(Pacifigorgia .).” Species?
- Line 251: “ventured to sea fans with different color to their own.” Awkward phrasing, please change.

I hope my suggestions can help the authors to improve the manuscript in order to achieve a publishable paper.


Joana Robalo
([email protected])

Reviewer 2 ·

Basic reporting

I consider the description of the game would be more appropriate for an educational journal.

Experimental design

The study aimed to describe host preference, habitat use and phylogenetic relationships among different color variants of egg-cowries. However, host preference is not described and habitat use is barely summarized (ie octocorals apparently served as habitat, feeding ground, site for reproductive aggregation and egg oviposition). Natural history observations of egg-cowries seemed rather anecdotal, lacking quantification in the form of number of octocoral colonies examined, number of egg-cowries per colony, predation effect, proportion of each of the two types of mimicry mismatches observed (shell color and oviposition color), etc; thus it is difficult to follow or to assess inferences about background matching being under strong selection pressure.

Validity of the findings

Lack of description on the methods, number of replicates or observations made about the natural history observations, or ecological results.

Additional comments

This study examines the phylogenetic relationships among color variants of egg-cowries using DNA sequencing of mitochondrial genes COI and 16S rDNA. For egg-cowries living in more than 8 species of Pacifigorgia-Eugorgia octocorals, mitochondrial divergence was negligible between two species, and between different color variants and morphologies within a species. I consider that these results suggesting that there is cross-fertilization between at least two species of egg-cowries are very valuable. Also, egg-cowries living in Leptogorgia octocorals form a different clade from those living in Pacifigorgia-Eugorgia colonies.

---

## Round 0.2 · Minor Revisions

The reviewers had a positive assessment of your resubmitted manuscript. However there are still some issues that need to be addressed. The language and presentation of the "board game" need to be revised. I also encourage the authors to pay close attention to those areas in the paper that have been identified by reviewer #2 as vague result descriptions and vague natural history observations.

·

Basic reporting

The manuscript “The Masquerade Game: marine mimicry adaptation between egg-cowries and octocorals” (#8738), written by Sánchez and colleagues aims to describe the marine mimicry adaptation present in egg-cowries (genus Simnia/Simnialena) and octocorals (genus Pacifigorgia/Eugorgia/Leptogorgia) from Tropical Eastern Pacific (TEP) in Colombia (Malpelo Island and Cabo Corrientes).

Experimental design

In my view this new version of the manuscript corrects the problems and answer most of my major and smaller concerns.

Validity of the findings

Nevertheless, in my opinion, the game, although very interesting, should be entirely removed from the paper and kept only as supplementary material. I understand the motives raised by the authors and I'm very keen of new forms of outreach, but I would be more favourable to present it out of the scientific paper. In the end, I leave that decision to the editor.

Additional comments

I hope my suggestions did help the authors to improve the manuscript.

Joana Robalo
([email protected])

Reviewer 2 ·

Basic reporting

I consider that the development of the board game is really out of the scope of this journal. I suggest that the authors submit the development of the board game to the “Science and Medical Education Overlay”, which is a subset of the peer-reviewed articles and rapid preprints found in PeerJ and PeerJ Preprints, respectively.
If a decision is made in favor to include this section, the language and style used (L231-277, Table 2, L370-388, L466-488) need to be improved and streamlined to be concordant to the rest of the text.
A great proportion of the discussion about the board game is focused to justify its inclusion in the manuscript, and should be avoided. For example, L462-464 “Moreover, the influence of the outreach activities can be boosted by the international distribution channels of scientific literature.” This sentence is really confusing. It can be interpreted as if educational journals are not as scientific as PeerJ or as if research in education, which has many dedicated journals, is not scientific literature. To avoid this kind of misinterpretation, the idea probably needs re-wording.

Experimental design

The M&M section “Host preference observations” has incorporated detailed information about sites, years, annual or biannual observation, etc. However, all those details provided seem unnecessary as there is no quantitative information that refers to those factors in the Result section. For example, there are no results showing host preference of any kind that include years, sites, or number of host colonies per site, depth interval or permanent transects, or number of egg-cowries per colony (or per transect, site, year, season of the year).

Validity of the findings

Detailed observations (as explained in the M&M section “Host preference observations”) are clearly absent in the Results section; thus, findings about this topic are extremely vague descriptions of the host preference and other natural history observations. In the following examples, I put in [brackets] vague descriptions:

L291-294:” Detailed observations at Malpelo Island revealed that [about 10%] of the colonies have at least one egg-cowry and 4% can have an ovopostion, which corresponded to roughly 0.01 and 0.04 egg-cowries or ovopositions per square meter assuming a seafan density of 4 col m-2 (Sánchez et al., 2012).”
What is the number of colonies?

L303: “[Occasionally there were some mismatches] between shell color and coenenchyme color…”

L307: “…mismatches [were commonly seen] during group mating”

L307-309: “In Malpelo Island group ovopositions [were frequently observed] in July, March, and November, but it seems to occur all year around.” This particular statement is in clear contrast to what is explained in the M&M where it was emphasized that the observations in Malpelo Island were done bi-annually.

L309-310: “During such events we noted that [some females], ventured to sea fans with different color to their own”

L313-314: “We observed that snails with background mismatches [were commonly present] at the boundary…”

L345-347 “Though egg-cowries [were eventually seen] on sea fan colonies affected by these two stressors, [most of them] were observed on healthy ones. Thus, we could assume that egg-cowries avoid affected corals.”

All of the above examples indicate a lack of quantitative assessment of host preferences or the failure to show them in a way that is not anecdotal.
I understand that speculation is welcome in PeerJ but in the results section of this study, there is no quantification for background mismatching, neither for the cowries nor for their eggs; also, there is no quantification of predation. Although the anecdotal evidence (as shown in pictures of potential predators and mismatch) can help to assume that potential predators can take advantage of background mismatching; there is no supportive evidence to infer that “background matching is under strong selection pressure”. Thus, the following speculation seems excessive:
L318-320: “Given our observations of the egg-cowries natural history, we infer that background matching is under strong selection pressure because there are many potential natural predators that can take advantage of mismatches.”

Additional comments

Minor editorial suggestions:
L92:it would be useful to spell-out TEP as it is first mentioned in the Background section
L130-131: The text after the parenthesis seems unnecessary. I suggest: “…and conducted multiyear, detailed preference observations in one of the two locations (Malpelo Island).”
L178: substitute ‘where’ with ‘were’
L289: Fig 2 B-C does not illustrate what is claimed in the sentence as plates B & C are both for cowries on Leptogorgia species.
L292: correct ovoposition
L330, I suggest: Sea fans of these two species were as near as 1-5 mm from each other at their overlapping depth interval (4-10 m).

---

## Round 0.3 · accepted · Accept

The authors have addressed satisfactorily the comments and suggestions by all reviewers during several revision rounds.